# Theory of axo-axonic inhibition

**Romain Brette**📷*

Sorbonne Université, CNRS, Institute of Intelligent Systems and Robotics (ISIR), Paris, France

* romain.brette@inserm.fr

## Abstract

The axon initial segment of principal cells of the cortex and hippocampus is contacted by GABAergic interneurons called chandelier cells. The anatomy, as well as alterations in neurological diseases such as epilepsy, suggest that chandelier cells exert an important inhibitory control on action potential initiation. However, their functional role remains unclear, including whether their effect is indeed inhibitory or excitatory. One reason is that there is a relative gap in electrophysiological theory about the electrical effect of axo-axonic synapses. This contribution uses resistive coupling theory, a simplification of cable theory based on the observation that the small initial segment is resistively coupled to the large cell body acting as a current sink, to fill this gap. The main theoretical finding is that a synaptic input at the proximal axon shifts the action potential threshold by an amount equal to the product of synaptic conductance, driving force at threshold, and axial axonal resistance between the soma and either the synapse or of the middle of the initial segment, whichever is closer. The theory produces quantitative estimates useful to interpret experimental observations, and supports the idea that axo-axonic cells can potentially exert powerful inhibitory control on action potential initiation.

## Author summary

Chandelier cells form GABAergic synapses on the initial segment of pyramidal cells of the cortex and hippocampus. Despite their striking morphology and alterations in neurological diseases such as epilepsy, their functional role remains unclear. This study develops a quantitative theory to precisely assess the electrical impact of a synaptic input at the proximal axon. It shows that axo-axonic inhibition acts by shifting the action potential threshold proportionally to the synaptic conductance. This work underlines the role of chandelier cells in controlling action potential initiation and provides a quantitative tool to interpret experimental observations.

**Data availability statement:** Code can be found at https://github.com/romainbrette/theory-of-axoaxonic-inhibition.

**Funding:** This work was supported by Agence Nationale de la Recherche, https://anr.fr/ (ANR-20-CE30-0025-01, ANR-21-CE16-0013-02 and ANR-23-CE16-0020-02 to RB). The funder played no role in in the study design, data collection and analysis, decision to publish, or preparation of the manuscript.

**Competing interests:** The author has declared that no competing interests exist.

## Introduction

In most vertebrate neurons, action potentials (APs) initiate in a small axonal structure near the soma, the axon initial segment (AIS) [1,2]. In principal cells of the cortex and hippocampus, the AIS harbors $GABA_A$ receptors contacted by interneurons, most of which are axo-axonic cells (AACs), also called "chandelier" cells (about 60% of synapses on the proximal axon of visual cortical cells of mice [3]). Chandelier cells are electrically coupled fast-spiking inhibitory neurons [4,5] with a distinct morphology, forming cartridges along the AIS of principal cells [6–8].

This striking anatomy suggests that AACs exert control on AP initiation, and may have an important role in normal function, or in the maintenance of the excitatory-inhibitory balance. *In vivo*, AACs targeting principal cells of the hippocampus fire at the peak of theta oscillations, when principal cells are least active, and they stop firing during sharp waves [9,10]. AACs targeting principal cells of the primary visual cortex respond strongly to locomotion and visuomotor mismatch [11]. Alterations in axoaxonic inhibition have been reported in several neuropathologies, but their causal implication is unknown [12–15].

Despite these suggestive clues, the precise effect of axoaxonic inhibition on neural function remains unclear. In fact, there has been some controversy over whether it is indeed inhibitory or excitatory, as a number of *in vitro* studies indicate that chandelier cells have a depolarizing effect [16–18], while *in vivo* they have an inhibitory action on hippocampal pyramidal cells of adult mice [19]. This discrepancy might be related to developmental changes in the reversal potential of chloride, the ionic carrier of $GABA_A$ currents [20,21]. However, a recent *in vitro* study indicates that axoaxonic synapses still reduce excitability even when the reversal potential is substantially depolarized [22]. *In vivo*, blocking AAC activity globally does not produce obvious effects [23]. The interplay of axoaxonic synaptic currents with structural plasticity of the AIS is also unclear. In cultures, activity can induce a distal displacement of the AIS [24,25], but the axoaxonic synapses stay in place [20,26,27]. The electrophysiological significance of this fact is not entirely obvious.

What makes interpretation difficult is the relative lack of theory about axonal inhibition. Indeed, classical excitability theory has focused on isopotential and spatially homogeneous situations [28,29], while theoretical work on synaptic integration has addressed dendrites, notably by Rall [30]. Here I provide a biophysical understanding of the effect of AIS synapses on excitability, based on resistive coupling theory, a simplification of cable theory that applies to the particular geometrical configuration where APs initiate in a thin axon close to a much larger cell body, a situation typical of many vertebrate neurons [31–33]. The theory results in a simple finding: axoaxonic inhibition raises the somatic threshold for AP initiation in proportion of synaptic conductance, axosomatic coupling resistance and driving force at threshold: $\Delta V^* = g_s R_a(E_{GABA} - V^*)$ (where V* is threshold). Axoaxonic inhibition is most powerful on the second half on the AIS, and is more stable when it does not move along with the AIS.

## Results

### Resistive coupling between soma and proximal axon

In most vertebrate neurons, APs initiate on a thin axon, close to a much larger cell body. In this situation, the soma acts as a current sink for transmembrane currents in the proximal axon (Fig 1A). Indeed, the conductance towards the soma is much larger than towards the distal axon or through the axonal membrane. Thus, as is shown in Fig 1B in a simple biophysical model (dendrite, soma and axon of diameter 6 μm, 30 μm and 1 μm, respectively), a current injected at the

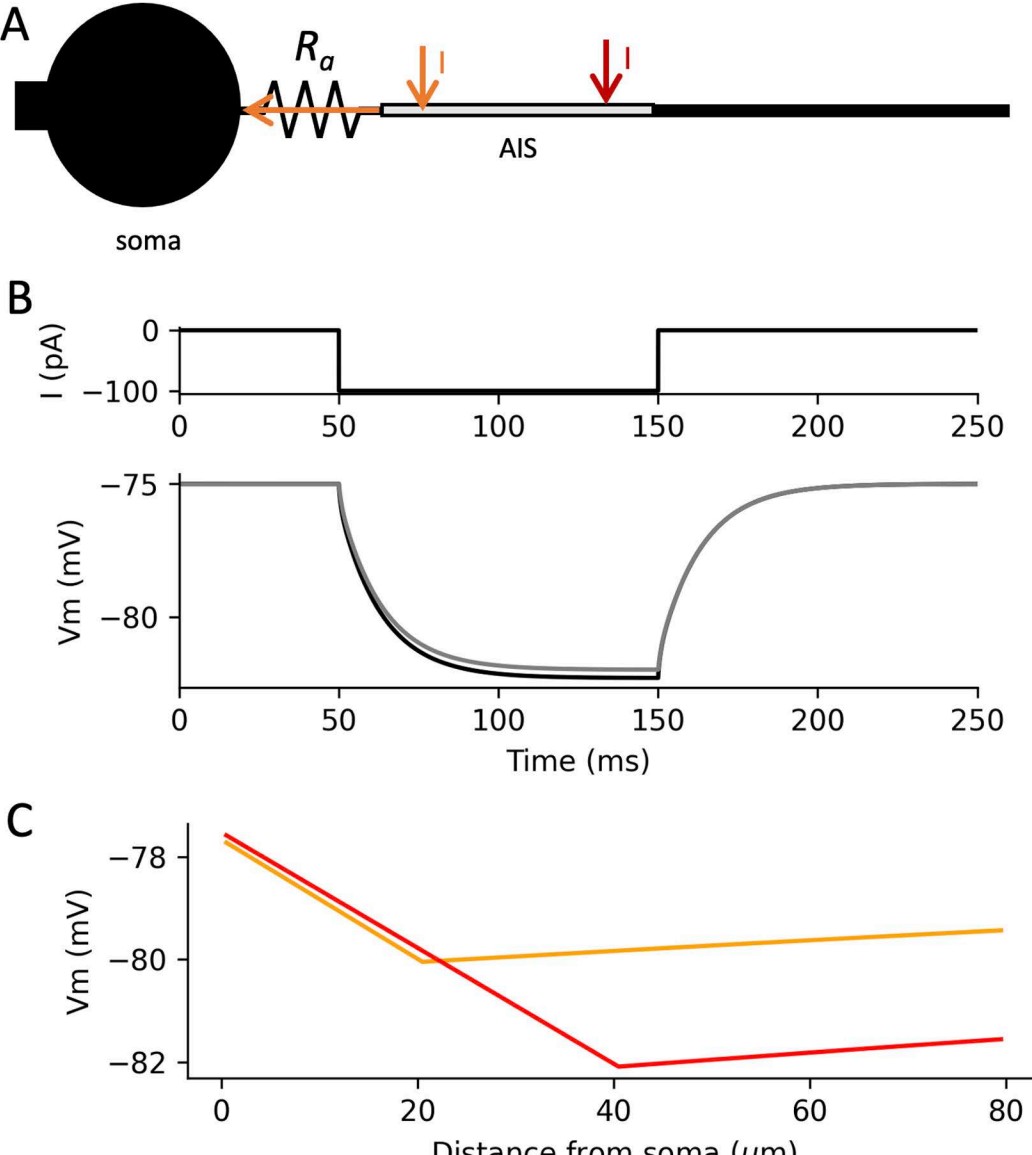

**Fig 1. Resistive coupling between soma and proximal axon, in a simple biophysical model. A,** A current I injected in the proximal soma (orange, 20 μm; red, 40 μm) flows mostly resistively towards the cell body. **B,** The somatic membrane potential in response to a current (top) injected in the soma (bottom, black) is essentially the same as when the current is injected in the proximal axon (grey, 40 μm). **C,** This results in a linear (ohmic) change in membrane potential between soma and injection site, with slope proportional to I (here shown at t = 5 ms).

proximal axon produces essentially the same voltage response at the soma than a current injected directly at the soma (grey, axonal injection 40 µm from the soma; black: somatic injection).

It follows that a current injected in the proximal axon flows mostly longitudinally towards the soma, and therefore resistively, thereby producing a nearly instantaneous ohmic voltage gradient between the soma and the current injection site (Fig 1C). This gradient follows Ohm's law, and is therefore equal to $R_a.I$, where $R_a$ is the axial (longitudinal) resistance between the two sites, on the order of 1 MΩ/µm in layer 5 pyramidal cells of the cortex (see Methods). Thus, with a cylindrical axon, the membrane potential varies linearly between soma and injection site, with a slope proportional to $I$. It follows that the same current injected 40 µm away from the soma (red) produces a local hyperpolarization that is twice larger than the same current injected 20 µm away (orange).

This phenomenon can be observed in dual soma-axon patch-clamp recordings of layer 5 cortical pyramidal cells, as shown in Fig 2 where the axonal electrode is placed 75 µm away from the soma (data reanalyzed from [34]). When a -50 pA current pulse is injected at the soma, it hyperpolarizes both the soma and the axon in the same way (left column), by about 5 mV. When the same current is injected in the axon, a~5 mV hyperpolarization also results in the soma, but an additional hyperpolarization is seen at the axonal site (~4 mV), which closely tracks the step current (right column). A

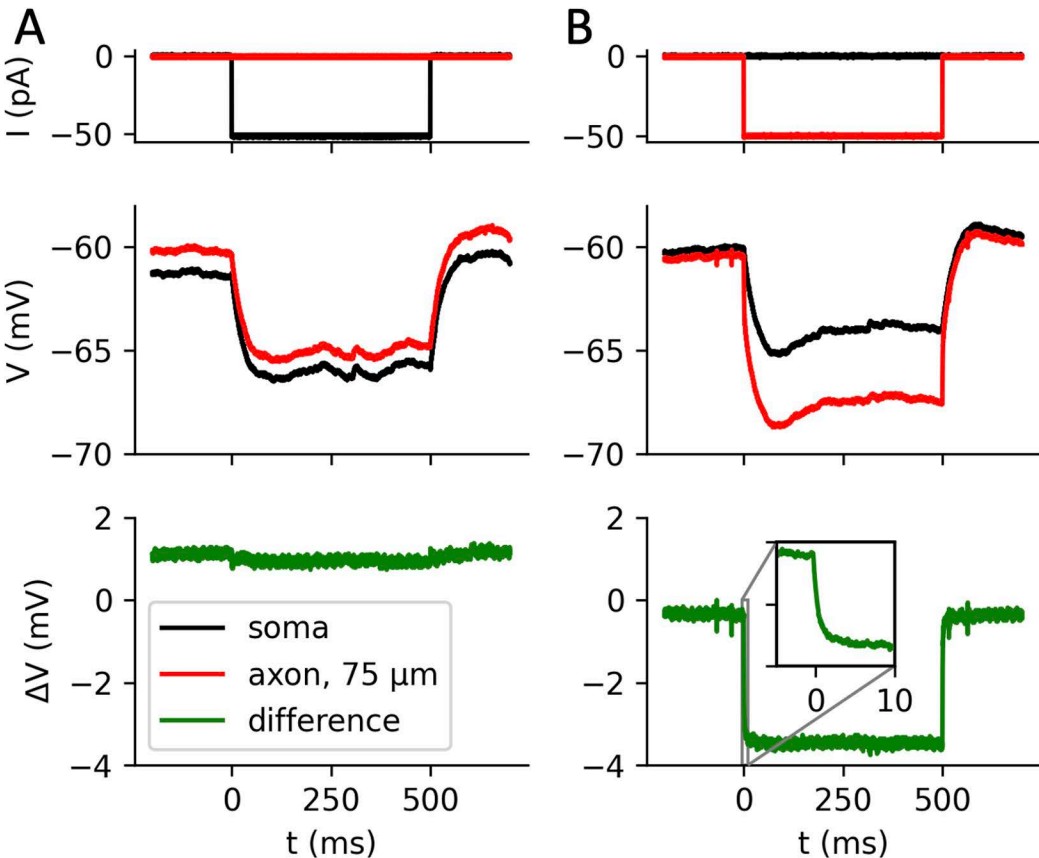

**Fig 2. Resistive coupling in simultaneous patch-clamp recordings of soma and axonal bleb of pyramidal cortical neurons (data from [ 34]).** *A, A negative current injected at the soma (top) results in the same hyperpolarization at the soma (middle, black) and proximal axon (red, 75 µm). The bottom trace shows the difference (green). B, The same current injected at the proximal axon (top), hyperpolarizes the axon (middle, red) more than the soma (black), with an additional ohmic component (green) due to the resistance between the soma and axonal injection site.*

zoom on the response onset shows that the establishment of this gradient is very fast, as expected from a resistive phenomenon (inset). The amplitude of the soma-axon voltage gradient matches the theoretical expectation based on Ohm's law (50 pA x 75 µm x 1 MΩ/µm = 3.75 mV).

Thus, the soma and proximal axon are resistively coupled. This simple resistive coupling greatly simplifies theoretical analysis. It allows quantitative understanding of the different factors influencing the initiation of APs [31,32,35], their backpropagation to the soma [36,37], and their extracellular signature [38]. Although relatively simple, the theory can be somewhat counter-intuitive. For example, a prediction of the theory is that increasing the axial resistance between the soma and AIS, as when the AIS shifts distally, makes the cell more excitable [31], and this prediction has been experimentally confirmed by pinching the proximal axon of pyramidal cells with glass pipettes [39].

We now examine the impact of synaptic currents at the proximal axon on AP initiation.

## Modulation of action potential initiation by axonal synaptic currents

A synaptic current on the proximal axon flows into the soma. Therefore, its effect on synaptic integration is essentially the same as a somatic current: a given current will produce the same somatic depolarization or hyperpolarization, whether injected at the AIS or at the soma (Fig 1B). However, there is an additional effect on AP initiation, as shown in Fig 3A on a simple biophysical model (see Methods, Numerical simulations). In this simulation, a hyperpolarizing current is injected in the middle of the AIS, extending from 5 to 35 µm from the soma, and an action potential is triggered by a depolarizing current pulse at the soma (1 nA, 5 ms). The somatic phase plot (dV/dt vs. V) shows the characteristic bimodal shape, where the first mode corresponds to the AP transmitted by the AIS and the second mode corresponds to the AP regenerated at the soma. The AP threshold (inflexion point of the first mode) increases with increasing synaptic current intensity. In contrast, when the same currents are injected at the soma, the AP threshold does not increase with current intensity (Fig 3B) – to be more accurate, the threshold slightly decreases, because of Nav channel inactivation.

Theoretically, the reason is simply that the axonal synaptic current $I$ (<0) produces a voltage gradient $R_a.I$ between soma and current injection site. The AP is triggered when the axonal potential $V + R_a I$ reaches axonal AP threshold $V_a^*$, therefore when the somatic potential $V$ reaches $V_a^* - R_a I$. It follows that the AP threshold at the soma is shifted by $-R_a I$. This theoretical analysis is based on a point AIS, but it matches results of numerical simulations on a spatially extended AIS, which indeed show an ohmic modulation of somatic AP threshold (Fig 3C). This modulation precisely matches the difference between somatic potential and average AIS potential, measured before the AP is triggered (Fig 3D).

As previously mentioned, the synaptic current may be depolarizing or hyperpolarizing, depending on whether the local membrane potential is above or below the reversal potential of chloride: $I = g(E_{GABA} - V)$, where $g$ is the synaptic conductance. The effect on AP threshold is therefore determined by the driving force at threshold: $I = g(E_{GABA} - V_a^*)$. Thus, in the biophysical model, the axonal synaptic current raises the threshold even when it is depolarizing at rest (Fig 3E). The threshold modulation reverses when $E_{GABA}$ is more depolarized than the AP threshold (dashed line). Note that at very depolarized values of $E_{GABA}$, the threshold modulation diverges from linearity: this is because of Nav channel inactivation, which raises the axonal threshold.

*In vitro* measurements indicate that, even when the synaptic current is depolarizing at rest ($E_{GABA} > V_{rest}$), $E_{GABA}$ remains about 10 mV below threshold [17], or is in the same range [21]. Substantial Nav channel inactivation would raise the threshold even more. Therefore, the generic effect of this axonal synaptic current should be to raise the somatic AP threshold, at all stages of development.

The synaptic conductance can also increase the axonal AP threshold by opposing the sodium conductance [32,40], i.e., by "shunting" the AP. It turns out that this effect is approximately equivalent to lowering the synaptic reversal potential by an amount $k_a$, the sodium channel activation slope (about 5 mV), which results in a simple formula for somatic threshold modulation: $\Delta V^* = g R_a (E_{GABA} - V^*)$, where $V^*$ is the somatic threshold in the absence of axonal synaptic current (see

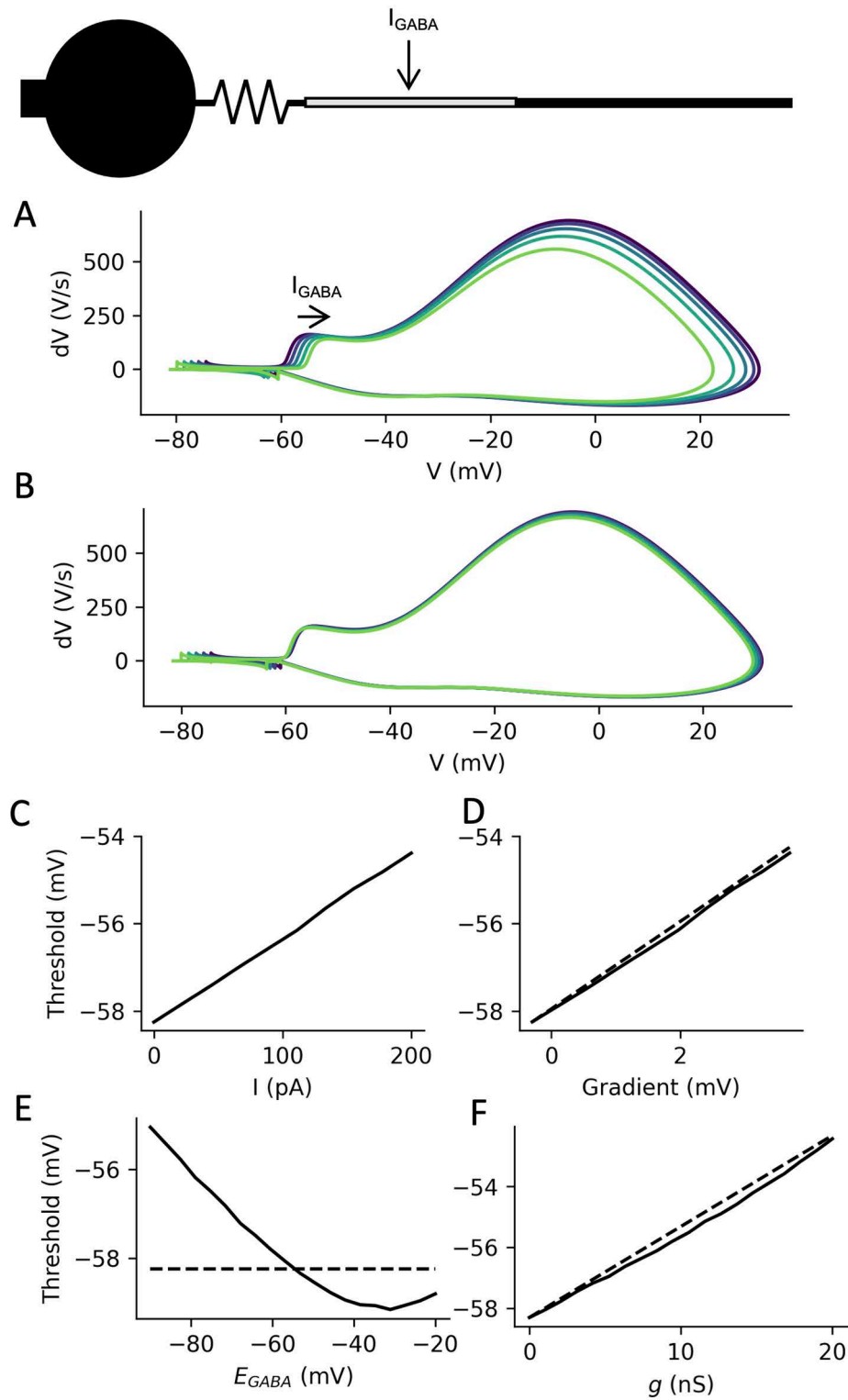

**Fig 3. Modulation of action potential initiation by axonal inhibition.** *A, Phase plots of somatic APs triggered by current pulses at the soma, when a negative current $I_{GABA}$ (0-200 pA) is injected in the middle of the AIS. B, Same as A, when the current is injected at the soma. C, Somatic AP threshold as a function of axonal current. D, Threshold as a function of the mean voltage gradient between soma and AIS, when the axonal current is varied between*

*0 and 200 pA (dashed: line of slope 1). E, With a synaptic current $I_{GABA} = g(E_{GABA}-V)$ (g = 5 nS), threshold as a function of synaptic reversal potential $E_{GABA}$. Dashed line: threshold with no synaptic current at the AIS. F, Threshold as a function of synaptic conductance g ($E_{GABA}$ = -70 mV). Dashed line: theoretical prediction.*

Methods, Theory). Therefore, threshold modulation is predicted to vary linearly with synaptic conductance. This theoretical formula appears to work very well in the biophysical model with spatially extended AIS, where the synaptic current is applied in the middle of the AIS (Fig 3F).

### Synapse position and structural plasticity of the AIS

How does synapse position along the axon impact AP threshold modulation? Suppose a current $I$ is injected on the proximal axon at distance $x$ from the soma. As we have seen, this produces an ohmic voltage gradient between soma and current injection site (Fig 1C). To a first approximation, the axial resistance of a piece of axon is proportional to its length L: $R_a = r_a L$, where $r_a$ is the axial resistance per unit length ($r_a \approx 1$ MΩ/μm in layer 5 pyramidal cells of the cortex; this value scales with axon diameter $d$ as $1/d^2$ [32]). Therefore, if the AIS is beyond the current injection site, then it gets hyperpolarized by an amount $\Delta V_{AIS} = r_a x I$, relative to the soma. If the AIS is between the soma and injection site, then it gets hyperpolarized by an amount $\Delta V_{AIS} = r_a x_{AIS} I$. In other words, the effect of an axonal synapse on threshold modulation should increase with its distance from the soma, but only up to the AIS.

 This theoretical analysis applies to a point AIS. When we measure threshold modulation as a function of synapse position in a biophysical model with a spatially extended AIS, we find that it increases approximately linearly with synapse position $x$ until (approximately) the middle of the AIS, with a slope as predicted from theory, then plateaus (Fig 4A and 4B). Therefore, synapses are most powerful when placed on the second half of the AIS, and placing them beyond the AIS does not increase their efficiency.

 What happens when the AIS moves with activity [41]? If the synapses are placed along the AIS and move together with it, then the impact of those synapses on threshold modulation should increase with distal shifts, as is seen in the biophysical model (Fig 4C). This change is more modest if synapses are fixed (Fig 4D).

 Empirically, it appears that most axo-axonic synapses are placed on the AIS [3,8,42] and remain in place when the AIS undergoes structural plasticity [20,26,27]. Our analysis indicates that this ensures a strong effect on AP threshold modulation while preserving some robustness to AIS displacements.

### Kinetics of threshold modulation

So far, we have examined the effect of constant axo-axonic inhibition. *In vitro*, the decay time of a postsynaptic current from a chandelier cell is about $\tau_s \approx 20$ ms in pyramidal cortical cells [43]. How does the threshold change over time when such a current is applied at the AIS? As we have seen, theory predicts a nearly instantaneous ohmic voltage gradient between the soma and the current injection site, following Ohm's law, and this is confirmed by simultaneous patch-clamp recordings (Fig 2B, inset). Accordingly, the AP threshold should closely track the applied current.

 In Fig 5, a 50 pA hyperpolarized current is applied at the middle of the AIS of the model, decaying with a 20 ms time constant (Fig 5A). At the soma, this results in an integrated membrane potential response peaking around t = 20 ms (Fig 5B). In contrast, the potential difference between soma and injection site closely tracks the applied current, as expected (Fig 5C). When we measure the AP threshold at different times by injecting a depolarizing current at the soma (here 0.7 nA), we find that the increase in threshold closely mirrors the voltage gradient and the applied current (Fig 5D). Thus, the time scale of threshold modulation is essentially that of the synaptic current.

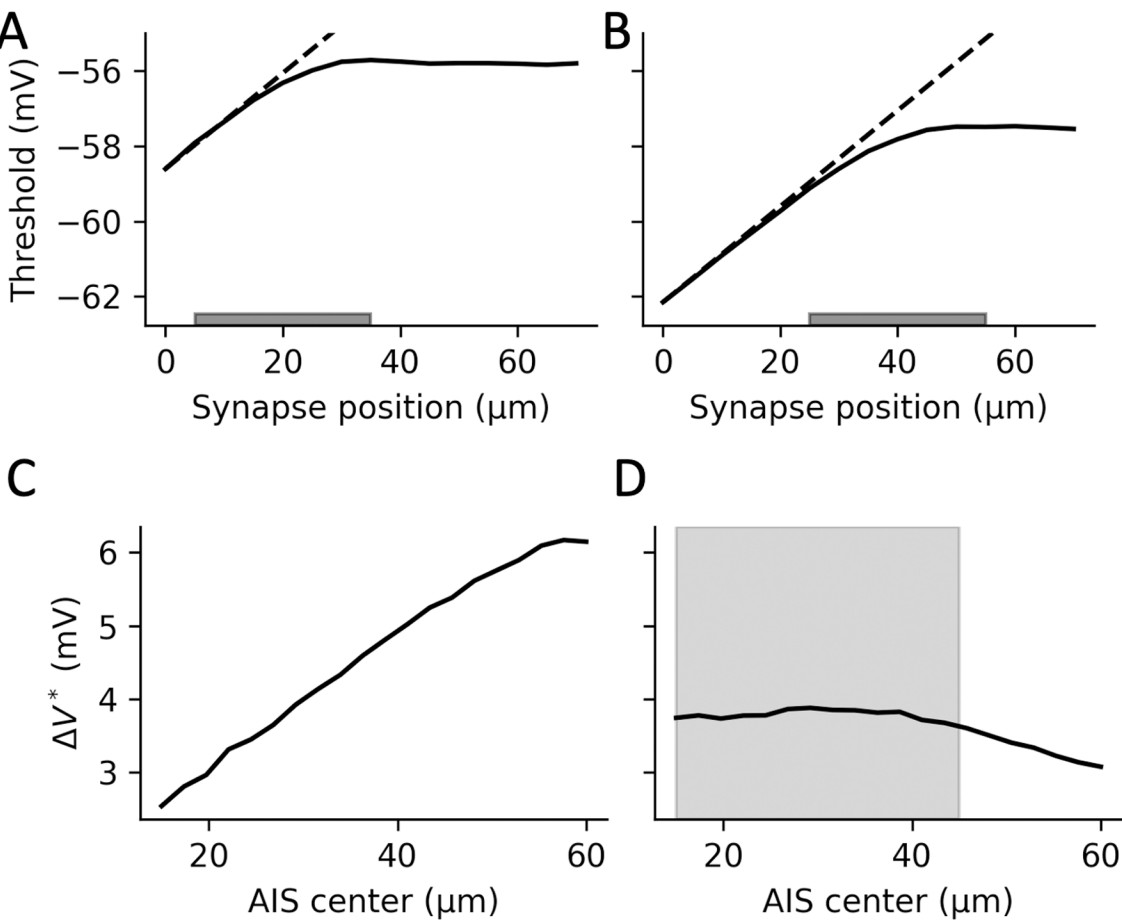

**Fig 4. *Effect of synapse and AIS position on threshold modulation.*** *A, Threshold as a function of synapse position, when the AIS extends from 5 to 35 μm from the soma ($I_{GABA}$ =-100 pA). Dashed line: theoretical prediction with a point AIS. B, Same as A, but the AIS extending from 25 to 55 μm from the soma. C, Change in threshold (relative to no axonal synaptic current) as a function of AIS center position, when synapses are distributed over the AIS and move with it ($E_{GABA}$ =-90 mV, g =5 nS). D, Same as C, but synapses are fixed between 15 and 45 μm from the soma (shaded area).*

## Discussion

### Summary

This theoretical analysis can be summarized by the following formula, which quantifies the change in AP threshold at the soma due to a synaptic current on the proximal axon:

$$\Delta V^* = g_s R_a (E_{GABA} - V^*)$$

where $R_a$ is the axial resistance from the soma to either the synapse or the middle of the AIS, whichever is closer; $g_s$ is synaptic conductance, $V^*$ is the somatic threshold without a synaptic current. From this analysis, we can conclude that the generic effect of GABAergic currents on the proximal axon is to raise the threshold for AP initiation, even when the current is depolarizing at rest.

This justifies the presumption that axoaxonic cells can potentially exert powerful control on AP initiation. The reason, as the formula emphasizes, is the relative electrical isolation of the AIS provided by the axosomatic coupling resistance

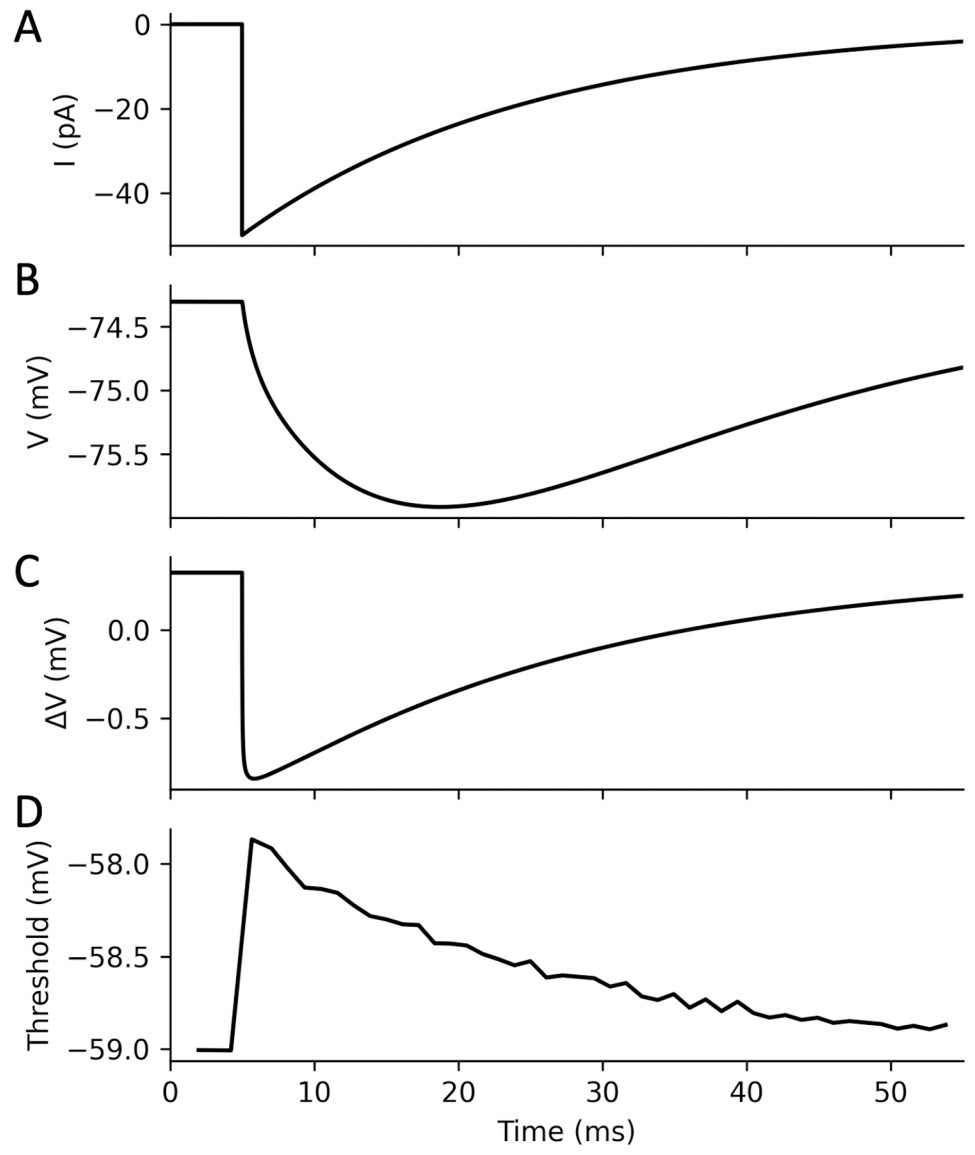

**Fig 5. Transient modulation of AP threshold by axo-axonic inhibition.** *A, A 50 pA hyperpolarizing current is applied at the middle of the AIS (decay time $\tau_s = 20\,ms$). B, Somatic membrane potential. C, Voltage gradient between soma and current injection site. D, Somatic threshold of AP initiation, measured by applying a 0.7 nA depolarizing current at the soma at different times (in steps of 1 ms).*

$R_a$. There is some uncertainty on the parameter values in the formula, and therefore on the magnitude of the threshold modulation exerted by axoaxonic cells, but it is possible to make some informed estimates. In layer 5 pyramidal cells of the cortex, the middle of the AIS is about 30 μm away from the soma [37], and axonal axial resistance is about 1 MΩ/μm in those neurons (see Methods), based on dual soma-bleb patch-clamp recordings [34]. Thus, a reasonable estimate is $R_a \approx 30$ MΩ. Tamás and Szabadics [43] report a peak postsynaptic current of about 50 pA for a single AAC targeting a layer 4 pyramidal cell held at -50 mV, in P24 rats. At that age, $E_{GABA} \approx -70$ mV [21]. Therefore, the conductance is $g \approx 2.5$ nS. Assuming a threshold of $V^* \approx -55$ mV, we obtain $\Delta V^* \approx 1.1$ mV for this single spike received from an AAC. If the synaptic conductance does not change with age, then at a later age when $E_{GABA} \approx -90$ mV, we would get $\Delta V^* \approx 2.6$ mV.

How would this effect scale *in vivo* when several AACs fire repeatedly on the same AIS? Each AIS receives inputs from ~4 AACs, for a total of about 20 contacts [13,42]. Thus, a synchronous discharge from all AACs would raise the threshold by ~10 mV in an adult. When $n$ AACs fire tonically at frequency $F$, the mean synaptic conductance is $nFg\tau_s$, where $\tau_s$ is the synaptic current decay time, which is about $\tau_s \approx 20$ ms [43]. The mean firing rate of AACs during hippocampal theta waves is about 15 Hz [9]. This would make $\langle g \rangle \approx 3$ nS and therefore $\langle \Delta V^* \rangle \approx 3.15$ mV, with a periodic modulation from 0 to 6.3 mV. Finally, AACs are fast-spiking cells that can fire up to 100 Hz or more [44]. If all AACs on a given AIS fire at maximum rate, this would yield $\Delta V^* \approx 21$ mV, essentially blocking AP initiation, although this might be mitigated by synaptic depression. Of course, these are just orders of magnitude with substantial uncertainty.

## Perisomatic vs. axonal inhibition

A striking feature of chandelier cells is that they specifically target the AIS, while pyramidal cells also receive strong inhibition at the soma and proximal dendrites, in particular by basket cells. What is the difference between these two types of inhibition? As we have seen, a current has the same effect on the somatic potential whether it is injected at the proximal axon or at the soma, because the soma is a current sink (Fig 1B). The essential difference is that axonal inhibition additionally raises the AP threshold, whereas somatic inhibition does not (Fig 3). Quantitatively, a synaptic current $I$ changes the somatic potential by an amount $RI$, where $R$ is the input resistance at the soma (for a steady current). When injected at the AIS, this current also changes the AP threshold by an amount $-R_aI$. It follows that the distance to threshold increases by $RI$ with somatic inhibition vs. $(R+R_a)I$ with axonal inhibition. Thus, the relative extra effect of axonal inhibition is $R_a/R$.

In addition, the input resistance $R$ is much reduced *in vivo* by the higher synaptic conductance [45], while this does not affect the axial resistance $R_a$. To give an order of magnitude, the input resistance of pyramidal cortical cells is around 30 M$\Omega$ *in vivo* [46–48]; with $R_a = 30$ M$\Omega$ (estimate above), we find that the effect of axonal inhibition is about twice that of somatic inhibition (meaning that somatic inhibition must be twice stronger than axonal inhibition to produce the same effect).

Another important difference is that axo-axonic inhibition acts nearly instantaneously on excitability, unfiltered by somatic integration (Figs 5 and 2B, inset). Given that an AIS is contacted by just a few chandelier cells, this implies that axoaxonic inhibition likely takes the form of transient inhibitory periods rather than a constant inhibitory drive. This suggests in particular that chandelier cells may influence the timing of APs of pyramidal cells, perhaps contributing to the generation or transmission of fast rhythms such as gamma oscillations, or to their coupling to the theta rhythm.

Finally, besides excitability, axoaxonic inhibition also modulates the membrane potential of the AIS (Fig 1C), and therefore may interact with molecular processes in that compartment, in particular involved in plasticity (see below, *On the molecular organization of the AIS*).

## Previous computational studies

To my knowledge, this is the first theoretical analysis of axoaxonic inhibition, but it has been examined in model simulations. In particular, Douglas and Martin [49] simulated axoaxonic inhibition on a pyramidal cell and found only a modest effect. Unfortunately, code, equations and parameters were not disclosed. Nevertheless, the discrepancy is likely due to the properties of AP initiation in the model, as it appears that the same sodium channel conductance densities were used in the AIS and soma. A key feature of AP initiation in most vertebrate neurons is that it occurs first at the AIS, thanks to a higher Nav conductance density as well as electrical isolation from the soma [31,50]. The AP is then regenerated at the soma at a voltage about 30 mV higher than the AP threshold [36,51]. This is a critical prerequisite of resistive coupling theory [32], in particular for threshold modulation by currents at the AIS. Indeed, without it, the result of axoaxonic inhibition is that the AP initiates in the soma through the somatic Nav channels, and the inhibitory effect then only occurs through the soma acting as a current sink, i.e., as if inhibition were applied at the soma.

More recently, Schneider-Mizell et al. [3] simulated axoaxonic inhibition on morphologically reconstructed pyramidal cells, with parameters constrained by optimization on somatic voltage recordings, and found a stronger effect of axoaxonic inhibition relative to somatic inhibition, in some cases. Unfortunately, details of equations and parameter values were also not presented, and code was not available at the time of writing. Given that the model was optimized based on somatic recordings, it is unclear whether biophysical properties of the AIS were correctly captured – indeed, axonal AP initiation was explicitly enforced as an additional optimization criterion, which means that the chosen somatic features were not sufficient to capture AIS properties.

These remarks highlight the importance of the specific biophysical properties of the AIS for the regulation of AP initiation.

### On the molecular organization of the AIS

This study focused on the electrical aspects of axo-axonic inhibition, specifically on excitability. In recent years, a number of studies have uncovered the sophisticated molecular organization of the AIS [52]. The AIS has a high density of ionic channels, inserted at particular positions relative to the axonal scaffold. GABAergic synapses are found all over the AIS, and also to some extent beyond the AIS, but not so much before [13,26,27,42]. As we have seen, this corresponds to places where the inhibitory current has the greatest electrical impact on excitability.

At a finer scale, GABAergic receptors co-localize with clusters of Kv2.1 channels, Cav3.2 channels and cisternal organelles [53–55]. These organelles are calcium stores, unique to the AIS, which also harbor ryanodine receptors, responsible for calcium release. Action potentials are associated with a calcium influx in the AIS, part of which goes through Cav3 channels in microdomains [55]. Since calcium is known to modulate GABA-A receptors [56–59], as well as Kv2.1 channels [60,61], it seems likely that electrical activity modulates GABAergic currents in the AIS, in particular through Cav3.2 channels, possibly amplified by calcium-induced calcium release from the cisternal organelles – an amplification that can itself be regulated.

Finally, axo-axonic inhibition might interact electrically with the Cav3.2 channels of the AIS. Indeed, those channels mediate T-type currents, which have the particularity of being partially inactivated at rest, and deinactivated by prolonged hyperpolarization [62]. This means that, in adult animals, prolonged axoaxonic inhibition would plausibly deinactivate Cav3.2 channels, especially on the distal end of the AIS, where hyperpolarization is stronger. After inhibition is released, calcium currents would be transiently potentiated.

In conclusion, this biophysical analysis shows that that the known physiology and anatomy of AACs are compatible with a strong modulatory effect on AP initiation at a fine temporal scale, possibly interacting with molecular processes at the AIS, in particular involved in plasticity.

## Materials and methods

### Theory

Consider a point AIS on the proximal axon, at a distance $x$ from the soma. It is electrically connected to the soma through an axial resistance $R_a = r_a x$, where $r_a$ is the resistance per unit length (assuming a cylindrical axon). This resistance is related to axon diameter $d$ by the formula $r_a = \frac{4R_i}{\pi d^2}$, where $R_i$ is intracellular resistivity (of order 100 $\Omega$ .cm). For example, with $d = 1$ μm, $r_a = 1.3$ MΩ/μm. With $R_i = 100–150$ Ω.cm and $d = 1$-1.5 μm, the range is $r_a = 0.57$-1.91 MΩ/μm. It can also be calculated from simultaneous patch clamp measurements at the soma and axonal bleb, where a small negative pulse $I$ is injected at the bleb, with the formula $r_a = (\Delta V_{\text{bleb}} - \Delta V_{\text{soma}})/(xI)$. Using published data in layer 5 pyramidal cortical cells [63], we find $0.81 \pm 0.55$ MΩ/μm (mean $\pm$ standard deviation). This might be an underestimation, because of the leak caused by the bleb. Thus, the order of magnitude for those cells is about 1 MΩ/μm.

We then consider a synaptic current $I = g_s(E_{GABA} - V)$ placed on the AIS, where $g_s$ is the synaptic conductance. The sodium conductance is then opposed by two conductances in parallel: the axial conductance $R_a^{-1}$ and the synaptic conductance $g_s$. The AP threshold at the AIS varies with the logarithm of the total non-sodium conductance, as $k \log (R_a^{-1} + g_s)$, where $k \approx 5$ mV is the sodium channel activation slope [40,64]. Relative to the absence of synaptic current, the threshold shift is therefore $k \log (1 + g_s R_a)$. When $g_s$ is small compared to $R_a^{-1}$ (which is typically the case, see Discussion), this is approximately $kg R_a$ (Taylor expansion). Thus, the axonal AP threshold is $V_a^* + kg_s R_a$, where $V_a^*$ is the axonal threshold without synaptic conductance.

In addition, as this synaptic current flows mostly resistively towards the soma, it introduces a potential difference between soma and AIS equal to $-R_a I_s = R_a g_s (V_a - E_{GABA})$. Thus, an AP is initiated when $V_{soma} - R_a I_s = V_a^* + kg_s R_a$. Thus, the AP threshold at the soma is:

$$V_{soma}^* = V_a^* + kg_s R_a + R_a I_s = V_a^* + g_s R_a(k + V_a^* + kg_s R_a - E_{GABA})$$

Therefore, assuming again that $g_s R_a$ is small, the shift in somatic threshold relative to an absence of synaptic conductance is:

$$\Delta V^* \approx g_s R_a(V_a^* + k - E_{GABA})$$

As it turns out, the somatic threshold in the absence of synaptic conductance $V_{soma}^*$ is depolarized relative to the axonal threshold, by an amount $k$ ($\approx 5$ mV) [31], in agreement with measurements [51]. Therefore, the final formula is simply:

**Table 1.** *Parameters values of the biophysical model. Time constants corrected for temperature are indicated in brackets.*

| Passive properties | $R_m$ | 15 000 $\Omega.cm^2$ |
|---|---|---|
| | $E_L$ | -75 mV |
| | $R_i$ | 100 $\Omega.cm$ |
| | $C_m$ | 0.9 $\mu F/cm^2$ |
| Nav channels | $g_{Na}$, soma | 250 $S/m^2$ |
| | $g_{Na}$, dendrite and axon (non AIS) | 50 $S/m^2$ |
| | $g_{Na}$, AIS | 4000 $S/m^2$ |
| | $E_{Na}$ | 70 mV |
| | $V_m^{1/2}$, soma | -30 mV |
| | $V_h^{1/2}$, soma | -55 mV |
| | $V_m^{1/2}$, AIS | -35 mV |
| | $V_h^{1/2}$, AIS | -60 mV |
| | $k_m$ | 5 mV |
| | $k_h$ | 5 mV |
| | $\tau_m^*$ | 150 $\mu s$ (corrected: 54 $\mu s$) |
| | $\tau_h^*$ | 5 ms (corrected: 1.8 ms) |
| Kv1 channels | $g_K$, soma | 250 $S/m^2$ |
| | $g_K$, dendrite and axon (non AIS) | 50 $S/m^2$ |
| | $g_K$, AIS | 1500 $S/m^2$ |
| | $E_K$ | -90 mV |
| | $V_n^{1/2}$ | -70 mV |
| | $k_n$ | 20 mV |
| | $\tau_n^*$ | 1 ms |

$$\Delta V^* = g_s R_a (V^*_{soma} - E_{GABA})$$

The theory with a spatially extended AIS is more complex, but we previously found that the threshold in an extended AIS is close to the value obtained in a point AIS with the same total conductances, placed at the middle of the extended AIS [32].

## Numerical simulations

Models were simulated with the Brian2 simulator [65], using a 1 µs timestep. The biophysical model is described in [32]. Briefly, it consists of a 500 µm long axon (diameter 1 µm), a 30 µm soma and a 1 mm long dendrite (diameter 6 µm) with voltage-gated sodium and potassium channels. Parameter values are listed in Table 1.

AP threshold is measured as the inflexion point of the phase plot (maximum of d2V/dt²), which is similar to a threshold on dV/dt but more robust [35].

## Acknowledgments

I thank Marcel Stimberg for technical assistance.

## Author contributions

**Conceptualization:** Romain Brette.

**Formal analysis:** Romain Brette.

**Funding acquisition:** Romain Brette.

**Investigation:** Romain Brette.

**Methodology:** Romain Brette.

**Project administration:** Romain Brette.

**Software:** Romain Brette.

**Validation:** Romain Brette.

**Visualization:** Romain Brette.

**Writing – original draft:** Romain Brette.

**Writing – review & editing:** Romain Brette.

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
