## [Decision Letter · Decision Letter 0]

18 Mar 2025

PCOMPBIOL-D-25-00221

Theory of axo-axonic inhibition

PLOS Computational Biology

Dear Dr. Brette,

Thank you for submitting your manuscript to PLOS Computational Biology. After careful consideration, we feel that it has merit but does not fully meet PLOS Computational Biology's publication criteria as it currently stands. Therefore, we invite you to submit a revised version of the manuscript that addresses the points raised during the review process.

Please submit your revised manuscript within 30 days May 18 2025 11:59PM. If you will need more time than this to complete your revisions, please reply to this message or contact the journal office at ploscompbiol@plos.org. Please include the following items when submitting your revised manuscript:

We look forward to receiving your revised manuscript.

Kind regards,

Hermann Cuntz

Academic Editor

PLOS Computational Biology

Marieke van Vugt

Section Editor

PLOS Computational Biology

**Journal Requirements:**

3) Please amend your detailed Financial Disclosure statement. This is published with the article. It must therefore be completed in full sentences and contain the exact wording you wish to be published.

**Reviewers' comments:**

Reviewer's Responses to Questions

Reviewer #1: This paper offers a tightly focused and important analysis examining the predictions of cable theory regarding GABAergic synaptic input on an axon initial segment. As described in the succinct but thorough introduction, the functional contributions of AIS input (particularly via chandelier cells) , the effect of such synapses have been surprisingly unclear despite the intuition that they afford powerful inhibition. This work grounds this intuition in a straight-forward, clearly written, and effective analysis, showing that this can occur via an effective increase in firing threshold that could strongly suppress the activity of target pyramidal cells under biologically feasible conditions. This fills a key gap in the literature, and provides useful baseline predictions for the effect of chandelier cells on neuronal networks, even if the unique biophysics of the AIS might complicate the real biology.

Major Suggestions:

* One of the most striking examples of inhibitory specificity is that chandelier cells _only_ target the axon initial segments. Given that somatic inhibition can also strongly suppress the activity of target pyramidal cells, it would be useful to thoroughly and quantitatively compare the effect of perisomatic inhibition with that of axonal inhibition. This contrast is briefly mentioned to in the text around line 139/140, but an explicit comparison would be valuable in trying to distinguish the roles of these two non-overlapping categories of inhibition, and this paper is well-suited to bring both categories of inhibitory targeting into the same analytical framework.

* The biophysics of the AIS is complex (e.g. the very nice recent review by Jenkins and Bender, Axon Initial Segment Structure and Function in Health and Disease, 2025 Physiological Reviews), and there are structural hints that Chandelier cells react to this. In particular, chandelier synapses cluster onto cisternal organelles associated with ryanodine receptors suggesting some connection with calcium-dependent calcium release. There are also important differences in sodium and potassium channels compared to elsewhere on the neuronal surface. Notably, GABAergic input at the AIS seems focused on this biophysically distinct region and not just the initial span of AIS. The theory presented here is based in cable theory and offers a useful baseline model that has been generally lacking from the literature, but it would be useful to add some discussion of how AIS-specific biophysics might act on top of the phenomenon analyzed here.

* There are some computational attempts to study AIS inhibition that are not discussed. Douglas and Martin 1990 (Control of Neuronal Output by Inhibition at the Axon Initial Segment) examined a similar geometry through simulation of Hodgkin-Huxley dynamics and came to a somewhat similar conclusion that AIS inhibition acts via controlling spike threshold. This study should be included in the otherwise excellent literature review in this paper, and it would be interesting to compare the results here with those found by Douglas and Martin. It would be instructive to compare the results here with those of this previous study, particularly as that study found it difficult for inhibition to overcome strong excitation, which is at least naively opposite to the estimates presented in the discussion (line 240–241). Similarly, Schneider-Mizell et al. 2021 included nonlinear biophysical simulations of inhibition at the AIS and compared to somatic inhibition, and could be included in this discussion.

Minor points:

* Line 96: A citation for the axial resistance of layer 5 pyramidal cells would be good.

Reviewer #2: This is an excellent paper. Dr. Brette presents an analysis of the effect of GABAergic synaptic currents occurring at the axon initial segment, as occurs with axon-axonic synapses. The analysis is presented in a highly accessible manner that nicely communicates the concepts as well as the results to a general reader. The presentation nicely combines a simple conceptual view of voltage drops across the axon coupling the AIS to the soma with full simulations of neuronal firing. The paper is exceptionally well-written. It is concise and clear in both the presentation of results and in the discussion.

I have only two suggestions for the author to consider.

One is an explicit statement about how the effect of a given synaptic conductance at the AIS would differ from the same conductance on the soma. This is addressed indirectly in lines 185-194 in the context of analyzing synapses position along the AIS but a sentence explicitly comparing to a soma location might be useful.

The second is a more complete consideration of the transient nature of the GABAergic currents. The analysis is of the steady-state situation but this is a simplification since the time-course of the synaptic conductance from a single firing of a presynaptic GABAergic is not long relative to the time course of capacitative charging. This point is alluded to in lines 236-240 but perhaps a more explicit statement would be helpful to the reader.

I would leave both of these suggestions to be considered or rejected as the author feels is best.

**Have the authors made all data and (if applicable) computational code underlying the findings in their manuscript fully available?**

Reviewer #1: Yes

Reviewer #2: Yes

PLOS authors have the option to publish the peer review history of their article (what does this mean? ). If published, this will include your full peer review and any attached files.

**Do you want your identity to be public for this peer review?** For information about this choice, including consent withdrawal, please see our Privacy Policy .

Reviewer #1: No

Reviewer #2: No

**Figure resubmission:**
---

## [Editor Report · Decision Letter 1]

11 Apr 2025

Dear Dr. Brette,

We are pleased to inform you that your manuscript 'Theory of axo-axonic inhibition' has been provisionally accepted for publication in PLOS Computational Biology.

Best regards,

Hermann Cuntz

Academic Editor

PLOS Computational Biology

Marieke van Vugt

Section Editor

PLOS Computational Biology

---

## [Editor Report · Acceptance letter]

PCOMPBIOL-D-25-00221R1

Theory of axo-axonic inhibition

Dear Dr Brette,

I am pleased to inform you that your manuscript has been formally accepted for publication in PLOS Computational Biology. Your manuscript is now with our production department and you will be notified of the publication date in due course.

With kind regards,

Anita Estes
